# Extensive qPCR analysis reveals altered gene expression in middle ear mucosa from cholesteatoma patients

Cecilia Drakskog[1], Nele de Klerk[1], Johanna Westerberg[2,3], Elina Mäki-Torkko[2,3,4], Susanna Kumlien Georén[1], Lars Olaf Cardell[1,5]*

1 The Division of ENT Diseases, CLINTEC, Karolinska Institutet, Stockholm, Sweden, 2 Division of Neuro and Inflammation Science, Department of Clinical and Experimental Medicine, Linköping University, Linköping, Sweden, 3 Department of Otorhinolaryngology in Linköping, Anaesthetics, Operations and Specialty Surgery Centre, Region Östergötland, Linköping, Sweden, 4 Faculty of Medicine and Health, Örebro University, Örebro, Sweden, 5 Department of ENT Diseases, Karolinska University Hospital, Stockholm, Sweden

☯ These authors contributed equally to this work.
* lars-olaf.cardell@ki.se

## Abstract

The middle ear is a small and hard to reach compartment, limiting the amount of tissue that can be extracted and the possibilities for studying the molecular mechanisms behind diseases like cholesteatoma. In this paper 14 reference gene candidates were evaluated in the middle ear mucosa of cholesteatoma patients and two different control tissues. *ACTB* and *GAPDH* were shown to be the optimal genes for the normalisation of target gene expression when investigating middle ear mucosa in multiplex qPCR analysis. Validation of reference genes using *c-MYC* expression confirmed the suitability of *ACTB* and *GAPDH* as reference genes and showed an upregulation of *c-MYC* in middle ear mucosa during cholesteatoma. The occurrence of participants of the innate immunity, *TLR2* and *TLR4*, were analysed in order to compare healthy middle ear mucosa to cholesteatoma. Analysis of *TLR2* and *TLR4* showed variable results depending on control tissue used, highlighting the importance of selecting relevant control tissue when investigating causes for disease. It is our belief that a consensus regarding reference genes and control tissue will contribute to the comparability and reproducibility of studies within the field.

## Introduction

The middle ear, also known as the tympanic cavity (TC), communicates with the mastoid air cells via the mastoid antrum (MA). The mucosa of the middle ear is continuous with the mucosa in the mastoid, consisting of a single layer of flattened to cuboidal respiratory epithelium [1]. Cholesteatomas affecting the middle ear and temporal bone can be acquired or congenital and consist of a keratinizing stratified squamous epithelium (matrix), with a cystic content of desquamated keratin debris. The cholesteatoma is surrounded by sub-epithelial tissue (peri-matrix) [2]. The pathogenesis of the acquired cholesteatoma is not fully understood

**Data Availability Statement:** All raw data files are available from the Zenodo database (DOI 10.5281/zenodo.3991972).

**Funding:** The study was financed by the Swedish research council. The funders had no role in study design, data collection and analysis, decision to publish, or preparation of the manuscript. No author received salary directly from the funders.

**Competing interests:** The authors have declared that no competing interests exist.

and cannot be explained by a single theory on its own. A eustachian tube dysfunction is today the most accepted theory of the underlying cause to development of an eardrum retraction [3]. A subsequent formation of a non self-cleaning pocket of the eardrum, which defines a cholesteatoma, may develop [4]. Repeated infections and or inflammation seem to be one of the contributing conditions for the cholesteatoma process, and its expansion [5].

A widely accepted theory is the involvement of a Eustachian tube dysfunction causing a retracted tympanic membrane with the subsequent loss of its self-cleaning abilities. In ears affected by acute otitis media and otitis media with effusion structural changes can be seen in the tympanic membrane, which becomes predisposed to retractions [6, 7]. Extensive research has been done to find contributing factors to the pathogenesis of cholesteatoma at a molecular level [8].

In cholesteatoma, the keratinising squamous epithelium is believed to possess hyperproliferative characteristics [8], illustrated by the upregulation of the *c-MYC* gene within the cholesteatoma matrix [9]. Inflammation is often part of the pathology in cholesteatoma contributing to a local bone destruction [2, 10]. There is not seldom a concomitant infection of the medial part of the ear canal, which leads to a faster expansion of the disease, and the only treatment is surgery [4]. Palko *et al.* has also shown that *c-MYC* is upregulated in cholesteatoma samples [9].

There is a risk, especially in children, for recurrent cholesteatoma [11]. Since all cholesteatoma matrix is removed during surgery it is likely that contributing factors to disease development reside outside the cholesteatoma matrix itself. For other conditions, like nasal polyps, it has been shown that the apparent healthy tissue adjacent to the diseased tissue also contains disease characteristics on a molecular level and contributes to disease pathology [12, 13]. It is therefore of interest to examine the properties of the mucosa (other than perimatrix), in ears with cholesteatoma.

There is a relationship between a good immunologic response and the protection against chronic middle ear disease (CMED). Toll Like Receptors (*TLRs*), as part of the mucosal innate immune system, play an important role as the first line of defense against an invasion of pathogenic infectious agents [14, 15], but their possible role in the cholesteatoma pathogenesis still needs to be clarified [16–19].

In animal studies, *TLR2* respond to lipoproteins found on gram positive bacteria such as *Staphylococcus aureus* [20], whereas *TLR4* is seen to be essential for signaling of bacterial lipopolysaccharide, produced by Gram-negative organisms such as *Pseudomonas aeruginosa* [21, 22]. Both mentioned pathogens are frequently detected in infected cholesteatomas [23]. Leichtle et al. [24] concluded that induction of *TLR2* via *TLR4* signaling was of importance for the time resolution of non-typeable *H. influenzae*-induced otitis media. Jotic et al. [25] describe *TLR2* and *4* expression in chronic suppurative otitis media, as well as in cholesteatoma matrix and perimatrix, compared to normal mucosa.

Inflammation is often part of the pathology in cholesteatoma contributing to a local bone destruction [2, 10]. There is not seldom a concomitant infection of the medial part of the ear canal, which leads to a faster expansion of the disease, and the only treatment is surgery [4]. Interesting findings by Si et al. [19] and Jiang et al. [26] show a relationship between high expression of *TLR4* in cholesteatoma matrix, and local bone destruction and inflammatory response.

With a few exceptions, the most common tissue for investigation of *TLRs* in ears with cholesteatoma is the cholesteatoma matrix, with skin as control [18, 19, 25, 27, 28]. Si et al. [19] saw a higher number of *TLR4* positive cells in acquired cholesteatoma, compared to congenital. The authors also claim that a bacterial infection almost always is involved in acquired cholesteatoma. Jesic et al. [27] investigated *TLR* expression in both cholesteatoma perimatrix and

matrix. In children, a higher expression was seen of *TLR2* and *4* in cholesteatoma compared to skin. In adults *TLR2* and *4* expression in cholesteatoma was higher expressed compared to healthy mucosa. These findings are in line with Lee et al. [18] and Szczespański et al. [28] who also detected a higher expression of *TLR2* and *4* in cholesteatoma matrix compared to normal skin.

In addition to investigations above of the cholesteatoma matrix, contributive studies of *TLRs* in the middle ear milieu in CMED, are performed by Granath et al. [16] and Hirai et al. [17]. A down-regulation of *TLR4* was seen in middle ear mucosa from a group of patients with CMED, including cholesteatoma [16] and an up-regulation of *TLR2* and *4* in immunohisto-chemical analysis in tissue samples from five cholesteatoma patients, in comparison to normal mucosa [17], was detected respectively. Because of a small sample size, and as the data are addressed as preliminary, Hirai et al. concludes that further studies are needed to clarify the role of *TLRs* in CMED [17]. The middle ear scene is a site for pathology such as chronic suppurative otitis media, which predisposes for an ear drum retraction and subsequent CMED [3]. It is therefore of interest to add confirming data to deepen the understanding of the relationship between the innate immunity of the middle ear mucosa, in ears with cholesteatoma compared to healthy.

Two limiting factors for studies related to middle ear diseases is the amount of tissue that can be extracted without causing severe adverse side effects to the patient, as well as the restricted access to relevant healthy control tissue. The amount of tissue available is especially limited when studying seemingly healthy mucosa from TC, since only a very small amount can be safely removed from the patient without interfering with recovery. There is a risk for fibrous adhesions or even adhesive otitis media in cases of greater mucosal injuries in the middle ear [29]. Hence, there is a need for more refined methods when studying complex pathophysiology. Semi-quantitative PCR (qPCR) is a suitable method because multiple factors can be analysed using very small amounts of tissue with high sensitivity, specificity and reproducibility [30].

QPCR has been used in several studies related to diseases of the middle ear. However, there seems to be little consensus as to which reference genes to use for normalisation, leading to the use of several different genes without proper establishment of which genes are appropriate [9, 31–39]. To be able to obtain reliable results the use of stably expressed reference genes for normalisation is essential [40]. It has become clear over the last decade that there is not one universal reference gene suitable for all different tissues within the human body, rather there is a need to determine the best reference genes for each tissue type in order to obtain reliable data [40, 41]. Also, different conditions can affect expression, such as disease or culturing of cells. As no gene is truly stably expressed, it is recommended to use two or three reference genes instead of just one in order to minimize naturally occurring variation in expression [41]. Taking all this into account, appropriate reference genes need to be identified for each tissue studied. To our knowledge the best suited reference genes to use for normalisation in middle ear mucosa have not yet been determined.

Several algorithms have been developed to evaluate the stability of reference genes taking multiple factors into account, such as transcription efficiency and expression variation (Norm-Finder, geNorm and BestKeeper) [42–44]. As these software solutions have been made freely available by their developers and they facilitate the evaluation of stability, these algorithms provide useful tools for the identification of suitable reference genes. NormFinder applies mathematical modelling to estimate the expression variation of a gene and uses this to calculate a stability value where a lower stability value indicates a more stable gene [42]. geNorm compares the expression patterns of each gene to all other genes included and calculates a stability value based on similarity in expression pattern. The least stable gene is excluded one at a time

until the two most stable genes are left [43]. Like NormFinder, a lower stability value represents a more stable gene. BestKeeper combines the data from all tested genes into a BestKeeper Index and compares it to every single gene to calculate a correlation coefficient where a gene with a higher correlation coefficient is more stable. BestKeeper is limited to analyse just ten genes simultaneously [44].

The aims of this paper were to develop a stable qPCR method with validated reference genes and determination of a suitable control tissue for the investigation of different diseases within the middle ear. Using this method, we aimed to investigate whether the middle ear mucosa in ears with cholesteatoma has altered gene expression in relation to its control tissue by the examination of *TLR2*, *TLR4* and *c-MYC*.

## Results

### Patient data

A total of 162 samples were obtained and used for RNA purification. Due to low yield and purity a final set of 92 samples was used for cDNA synthesis and qPCR analysis. The MA samples were obtained from patients during cochlear implant (Cl) or translabyrinthine vestibular schwannoma surgery (n = 37). These included 17 men and 20 women with ages ranging from 16 to 85 (average 50 years). Healthy TC tissues were obtained from individuals undergoing translabyrinthine vestibular schwannoma surgery (n = 25). In this group the male/female ratio was 15/10 and the average age was 52 years (range 16–85 years). The cholesteatoma group included 30 patients of whom 17 were male and 13 female. The age ranged from 4 to 82 years with an average of 30 years. For more demographic data of the cholesteatoma group, see Table 1. The surgical method used is described by Westerberg et al. [45]. As our intention was to compare properties of the middle ear mucosa in healthy compared to cholesteatoma diseased ears, this was the chosen site for sampling of mucosa specimen. Caution was taken to avoid skin or cholesteatoma matrix in the samples.

### Gene selection and evaluation of primer efficiency

The 14 candidates for the identification of reference genes were chosen based on their frequent use as reference genes in literature and recommendations by primer manufacturers (Table 2). The primer amplification efficiencies for each gene were calculated using one primer set alone (singleplex) as well as in combination with a second candidate gene (multiplex) to determine possible competition effects on amplification. From the initial 14 candidate reference genes *TBP*, *TUB1A* and *TUBB2B* were excluded at this stage due to extremely low detectability (Ct > 35). For the remaining genes, of which the expression could be detected, the difference between singleplex and multiplex runs was minimal, indicating no or little competition during amplification. The primer efficiency for the candidate reference genes in multiplex assays ranged from 88.1% to 103.1% (Table 3).

### Expression patterns of candidate reference genes in middle ear mucosa

The expression of the eleven candidate reference genes was analysed in mucosal biopsies taken from the MA and the TC of healthy controls, and from the TC of cholesteatoma patients. A considerable variation was seen in expression levels between the different candidate reference genes, with mean Ct-values ranging from 17.74 for *GAPDH* in the mucosal biopsies from the TC in cholesteatoma patients to 27.70 for *CANX* in the MA from healthy controls (Table 4, Fig 1). However, the differences in expression levels between tissue groups for each gene were small; only the expression of *HPRT1*, *PGK1*, *PPIA*, *ATP5B* and *YWHAZ* was significantly

Table 1. Demographic data of cholesteatoma cases n = 30. If stapes was present a shorter 'slipper'-type ossiculo-plasty (PORP) was used, and if superstructures of stapes were absent a longer ossicle was employed (TORP).

| Variable | n (%) |
|---|---|
| Gender | |
| Male | 19 (63%) |
| Female | 11 (37%) |
| Diseased ear | |
| Right | 14 (47%) |
| Left | 16 (53%) |
| Age at surgery (years) | |
| Mean±SD | 30 ± 21 |
| Type of cholesteatoma | |
| Attic | 19 (63%) |
| Tensa | 8 (27%) |
| Combination of attic and tensa | 2 (7%) |
| indistinct | 1 (3%) |
| Surgical event | |
| Primary surgery | 29 (97%) |
| Revision | 1 (3%) |
| Auditory ossicular chain, peroperative findings | |
| Intact | 6 |
| Erosion of malleus | 2 |
| Erosion of incus | 14 |
| Erosion of malleus and incus | 2 |
| Erosion of incus and stapes | 2 |
| Erosion of malleus, incus and stapes | 4 |
| Surgical method | |
| Canal wall up (CWU) with obliteration* | 23 (87%) |
| Canal wall down (CWD) with obliteration** | 1 (3%) |
| Atticoantrotomy with obliteration | 6 (20%) |
| Reconstruction | |
| PORP | 19 (63%) |
| TORP | 6 (20%) |
| Intact ossicular chain*** | 5 (17%) |

*Comprising 10 cases of combined approach tympanoplasty (CAT).

**One patient had a complete sensorineural hearing loss preoperatively.

***In two cases the ossicular chain was not reconstructed as hearing restoration could not be expected, due to preoperative deafness, and stapes fixation, respectively.

higher in healthy MA compared to the mucosa from the middle ear in cholesteatoma patients (Fig 1). In all tissue groups *HPRT1*, *SDHA*, *GUSB* and *CANX* displayed the most variability in expression, whereas *ACTB*, *UBC*, *PPIA*, *ATP5B* and *GAPDH* were the least variable as shown by the whiskers of the box-plots and the standard deviations (Fig 1, Table 4).

## Analysis of gene transcription stability of candidate reference genes

Three Excel-based software solutions were used to evaluate the stability of the candidate reference genes in the different tissue groups: NormFinder [42], geNorm [43] and BestKeeper [44]. Because *CANX* showed large variability in expression (Fig 1) and was ranked among the least

**Table 2. Candidate reference genes.**

| Gene symbol | Gene name | Accession number | Function | Location | Pseudogenes |
|---|---|---|---|---|---|
| *ACTB* | actin beta | NM_001101 | Cytoskeletal structural protein | 7p22.1 | Yes |
| *SDHA* | succinate dehydrogenase complex flavoprotein subunit A | NM_004168 | Electron transporter in the mitochondrial respiratory chain and TCA cycle | 5p15.33 | Yes |
| *HPRT1* | hypoxanthine phosphoribosyltransferase 1 | NM_000194 | Purine nucleotide synthesis through purine salvage pathway | Xq26.2 | Yes |
| *UBC* | ubiquitin C | NM_021009 | Protein degradation | 12q24.31 | Yes |
| *PGK1* | phosphoglycerate kinase 1 | NM_000291 | Synthesis of 3-phosphoglycerate, cofactor for polymerase alpha, and angiogenesis in tumors | Xq21.1 | Yes |
| *PPIA* | peptidylpropyl isomerase A | NM_021130 | Protein folding | 7p13 | Yes |
| *ATP5B* | ATP synthase, H+ transporting. mitochondrial F1 complex. beta polypeptide | NM_001686 | ATP synthesis | 12q13.3 | Yes |
| *GUSB* | glucuronidase beta | NM_001284290 | Glycosaminoglycan hydrolosis | 7q11.21 | Yes |
| *GAPDH* | glyceraldehyde-3-phosphate dehydrogenase | NM_002046 | Oxidoreductase in glycolysis and gluconeogenesis | 12p13.31 | Yes |
| *YWHAZ* | tyrosine 3-monooxygenase/ tryptophan 5-monooxygenase activation protein zeta | NM_001135700 | Signal transduction by binding to phosphoserine-containing proteins | 8q22.3 | Yes |
| *CANX* | calnexin | NM_001024649 | Quality control of and facilitating protein folding and assembly | 5q35.3 | No |

stable genes in the NormFinder and geNorm analyses (Fig 2a–2f), this gene was omitted from analysis with BestKeeper. For TC samples from healthy controls and cholesteatoma patients, all three programs ranked *ACTB* and *GAPDH* among the most stable genes (Fig 2b, 2c, 2e, 2f, 2h and 2i). For MA biopsies from healthy controls the results were more variable between the different software solutions, although *ACTB* still ranked high in all three analyses. Other high scoring genes in this tissue group were *UBC* and *PPIA* (Fig 2a, 2d and 2g). Unexpectedly, Best-Keeper ranked *SDHA* most stable in MA tissue while this gene scored among the lowest in other analyses and tissues (Fig 2a–2i). Overall, *ACTB* is considered the most stable gene in all mucosal biopsies evaluated.

## Intergroup analysis of candidate reference gene stability

Besides the expression variation within one sample group, the expression variation between sample groups is determinative for the stability of a gene. NormFinder is the only software of the three used in this study that is able to determine this intergroup variation [42]. Since it was clear from the previous analysis that *SDHA*, *HPRT1*, *GUSB* and *CANX* were the most variably expressed, only the other seven genes were still considered possible suitable reference genes and were therefore included in the intergroup analysis. The intergroup variation should be as close to 0 as possible, because genes with values that deviate from 0 imply a systematically higher ($> 0$) or lower ($< 0$) expression compared to the other sample groups. Due to this skewing, differences in target gene expression between sample groups will either be over- or underestimated. The genes examined here showed minimal intergroup variation for all sample groups tested, presenting *GAPDH* top-ranked with the least variation (Fig 3a). Using the intragroup variation to calculate the confidence interval for the intergroup variation (displayed as vertical lines in Fig 3a) it seems that the most expression variation is seen in the healthy MA sample group and for the *PGK1* and *YWHAZ* genes. Taken together, the intergroup analysis ranks *GAPDH* as the most and *PGK1* and *YWHAZ* as the least stable genes (Fig 3a).

**Table 3. Primer and probe sequences.**

| Gene symbol | Primer and Probe sequences (5'– 3') | Tm (˚C) | Amplicon size (bp) | Oligo concentration (nM) | Efficiency (%) | Efficiency multiplex (%) |
|---|---|---|---|---|---|---|
| *ACTB* | F: CAAGATGAGATTGGCATGGC | 65.5 | 147 | 500/50 | 101.2 | 103.1 |
| | R: CACATTGTGAACTTTGGGGG | 65.1 | | 500/50 | | |
| | P: [HEX]TGACAGCAGTCGGTTGGAGCGAGCA[OQA] | 79.3 | | 200 | | |
| *SDHA* | F: TTGGACCTGGTTGTCTTTGG | 64.9 | 110 | 500 | 107.9 | 96.8 |
| | R: ATGACAGATTCTTCCCCAGC | 63.0 | | 500 | | |
| | P: [6FAM]AGCATCGAAGAGTCATGCAGGCCTGG[OQA] | 77.1 | | 200 | | |
| *HPRT1* | F: CTTTGCTTTCCTTGGTCAGG | 63.6 | 112 | 500 | 96.3 | 101.6 |
| | R: TCAAATCCAACAAAGTCTGGC | 63.9 | | 500 | | |
| | P: [6FAM]GCTTGCTGGTGAAAAGGACCCCACG[OQA] | 77.1 | | 200 | | |
| *UBC* | F: ACATCCAGAAAGAGTCCACC | 60.6 | 124 | 500 | 98.7 | 100.6 |
| | R: TTCTCAATGGTGTCACTCGG | 63.8 | | 500 | | |
| | P: [HEX]ACCTGGTGCTCCGTCTTAGAGGTGGGA[OQA] | 76.2 | | 200 | | |
| *PGK1* | F: TGGTGAAAGCCACTTCTAGG | 61.8 | 147 | 500 | 86.1 | 92.8 |
| | R: GGACTTTACCTTCCAGGAGC | 61.6 | | 500 | | |
| | P: [6FAM]ACTGCCACTTGCTGTGCCAAATGGAAC [OQA] | 76.4 | | 200 | | |
| *PPIA* | F: GACCCAACACAAATGGTTCC | 64.0 | 112 | 500 | 82.7 | 89.3 |
| | R: GCCTCCACAATATTCATGCC | 64.1 | | 500 | | |
| | P: [HEX]TGCCAAGACTGAGTGGTTGGATGGCA[OQA] | 77.6 | | 200 | | |
| *ATP5B* | F: CAGGCCTTCTATATGGTGGG | 63.1 | 98 | 500 | 95.5 | 94.8 |
| | R: GTACAGAGGACAAAGACCCC | 60.0 | | 500 | | |
| | P: [6FAM]GCTGTGGCAAAAGCTGATAAGCTGGCT [OQA] | 74.6 | | 200 | | |
| *GUSB* | F: ATTCAGAGCGAGTATGGAGC | 61.0 | 106 | 500 | 92.7 | 88.4 |
| | R: CCAGATGGTACTGCTCTAGC | 59.1 | | 500 | | |
| | P: [HEX]TGCAGGGTTTCACCAGGATCCACCTC[OQA] | 76.9 | | 200 | | |
| *GAPDH* | F: ACAACGAATTTGGCTACAGC | 61.7 | 121 | 500 | 90.4 | 88.1 |
| | R: AGTGAGGGTCTCTCTCTTCC | 59.0 | | 500 | | |
| | P: [HEX]ACCACCAGCCCCAGCAAGAGCACAA[OQA] | 78.5 | | 200 | | |
| *GAPDH* | F: ACAACGAATTTGGCTACAGC | 61.7 | 121 | 50 | 98.5 | 96.6 |
| | R: AGTGAGGGTCTCTCTCTTCC | 59.0 | | 50 | | |
| | P: [6FAM]ACCACCAGCCCCAGCAAGAGCACAA[OQA] | 78.5 | | 200 | | |
| *YWHAZ* | F: TAGCCTTCTGTCTTGTCACC | 59.5 | 129 | 500 | 94.5 | 92.8 |
| | F: ATAGTCTGTGGGATGCAAGC | 61.7 | | 500 | | |
| | P: [6FAM]AGGCCGCATGATCTTTCTGGCTCCA[OQA] | 77.4 | | 200 | | |
| *CANX* | F: GATGAGCCTGAGTACGTACC | 58.8 | 115 | 500 | 104.5 | 98.9 |
| | R: GAGCTGACTCACATCTAGGGP: | 59.0 | | 500 | | |
| | [6FAM]GGAGAATGGGAGGCTCCTCAGATTGCC[OQA] | 76.9 | | 200 | | |
| *c-MYC* | F: GACGCGGGGAGGCTATTC | 59.9 | 134 | 500 | 75.5 | 85.2 |
| | R: CGGGAGGCTGCTGGTTTT | 60.3 | | 500 | | |
| | P: [Cy5]CCGCTGCCAGGACCCGCTTCTCTGAAA[OQD] | 73.1 | | 200 | | |
| *TLR2* | F: TGTGGATGGTGTGGGTCTTG | 59.89 | 114 | 500 | 84.8 | 101.4 |
| | R: AAGATCCTGAGCTGCCCTTG | 59.74 | | 500 | | |
| | P: [Cy5] TCAGGCTTCTCTGTCTTGTGACCGCAATGG [OQD] | 70.94 | | 200 | | |

*(Continued)*

**Table 3.** (Continued)

| Gene symbol | Primer and Probe sequences (5'– 3') | Tm (°C) | Amplicon size (bp) | Oligo concentration (nM) | Efficiency (%) | Efficiency multiplex (%) |
|---|---|---|---|---|---|---|
| *TLR4* | F: CAGAGTTTCCTGCAATGGATCAAG | 60.14 | 86 | 500 | 96.1 | 81.5 |
| | R: TGCTTATCTGAAGGTGTTGCACAT | 61.05 | | 500 | | |
| | P: [Cy5] AGAGGCAGCTCTTGGTGGAAGTTGAACGA [OQD] | 70.42 | | 200 | | |

F: forward primer, R: reverse primer, P: dual-labeled probe, Tm: melting temperature, bp: base pairs, [HEX]: hexachloro-6-carboxyfluorescein, [6FAM]: 6-carboxyfluorescein, [Cy5]: Cyanine 5, [OQA]: Onyx Quencher™ A [OQD]: Onyx Quencher™ D.

## Summarized ranking of candidate reference genes

Taking together the expression patterns, the transcription stabilities and the intergroup variation it is clear that *ACTB* and *GAPDH* are the most suitable reference genes for analysing middle ear tissue, closely followed by *UBC*, *PPIA* and *ATP5B*. Reference genes that should not be used are *SDHA*, *HPRT1*, *GUSB* and *CANX* (Fig 3b).

## Reference gene validation and analysis of c-MYC expression

In order to validate the choice of suitable reference genes, the highest ranked and lowest ranked genes were used for normalisation of the expression of *c-MYC*. *C-MYC* is a proto-oncogene with a well-known role in tumorigenesis [46]. As mentioned above, *c-MYC* is upregulated in cholesteatoma samples compared to retroauricular skin samples [9]. Since *PPIA* was used as a reference gene in the mentioned study, this gene was included in the validation analysis in addition to *ACTB+GAPDH* (highest ranked) and *SDHA+GUSB* (lowest ranked). First, the transcription efficiency of the *c-MYC* primer pair was determined and shown to be 85.2% in the multiplex assays (Table 3). When comparing the relative expression of *c-MYC* normalised to the different reference genes there is an obvious difference in expression variation. The variation in relative expression levels of *c-MYC* is 10 857 fold (min 0.1400, max 1520) using *SDHA* and *GUSB* as reference genes, 8818 fold (min 0.0001, max 0.8818) using *PPIA*, while only 2003 fold (min 0.00014, max 0.28046) using *ACTB* and *GAPDH* (Fig 4a–4c). This shows that the stability of the reference genes affects the variability of detection of the target gene. A higher variability can in turn compromise the detection of differences in target gene expression between groups. Further indication for the suitability of the reference genes is the target

**Table 4. Ct values.**

| | | *ACTB* | *SDHA* | *HPRT1* | *UBC* | *PGK1* | *PPIA* | *ATP5B* | *GUSB* | *GAPDH* | *YWHAZ* | *CANX* |
|---|---|---|---|---|---|---|---|---|---|---|---|---|
| MA, healthy | *Mean* | 19.90 | 25.01 | 27.09 | 19.33 | 23.71 | 21.77 | 20.29 | 26.74 | 18.78 | 25.38 | 27.70 |
| | *SD* | 1.27 | 2.60 | 2.02 | 1.04 | 1.61 | 1.27 | 0.82 | 2.16 | 0.86 | 1.77 | 2.89 |
| TC, healthy | *Mean* | 19.83 | 23.16 | 25.88 | 18.68 | 22.76 | 21.33 | 19.61 | 25.75 | 18.34 | 24.45 | 26.39 |
| | *SD* | 0.75 | 1.17 | 1.03 | 0.71 | 0.89 | 0.69 | 0.61 | 1.72 | 0.67 | 0.99 | 2.01 |
| TC, cholesteatoma | *Mean* | 18.97 | 23.96 | 25.69 | 18.69 | 22.55 | 20.60 | 19.74 | 25.84 | 17.74 | 24.08 | 25.99 |
| | *SD* | 1.08 | 1.55 | 1.01 | 1.00 | 1.21 | 0.74 | 0.84 | 1.95 | 1.03 | 1.19 | 1.74 |
| All tissue groups | *Mean* | 19.57 | 24.13 | 26.29 | 18.94 | 23.06 | 21.26 | 19.92 | 26.16 | 18.31 | 24.69 | 26.76 |
| | *SD* | 1.12 | 1.93 | 1.49 | 0.96 | 1.33 | 1.00 | 0.79 | 1.98 | 0.94 | 1.44 | 2.31 |

Geometric mean ± standard deviation (SD).

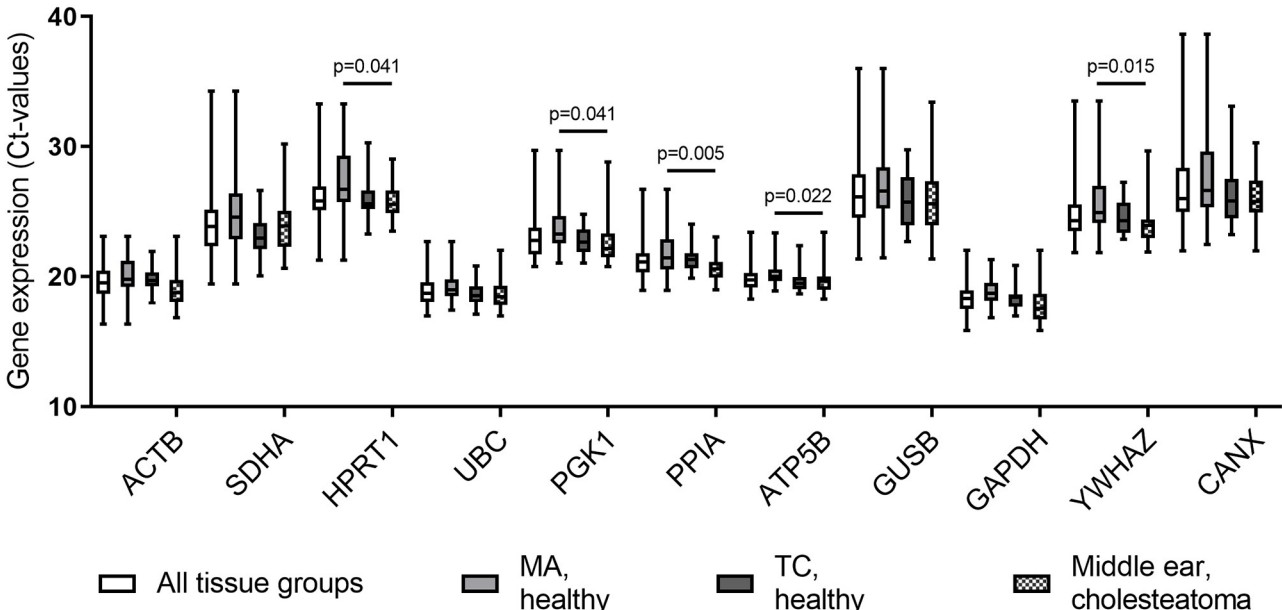

**Fig 1. Ct-value distribution of reference genes.** Box and whisker plots showing mRNA expression levels depicted as raw Ct-values for all tissue groups (n = 87), mastoid antrum (MA) mucosa from healthy controls (n = 35), tympanic cavity (TC) mucosa from healthy controls (n = 25) and from cholesteatoma patients (n = 28). Boxes show the 25th, 50th and 75th percentiles; whiskers depict the minimum and maximum Ct-values for each reference gene. For data sets considered Gaussian distributed (*ACTB* and *GAPDH*) one-way ANOVA with Holm-Sidak post-test was used for statistical analysis. For non-Gaussian distributed data sets one-way Kruskal-Wallis test with Dunn's post-test for multiple comparisons were performed. For groups where statistical difference was observed the p-values are presented.

gene expression in the control tissues. Because both control tissues are healthy tissues no difference in *c-MYC* expression is expected. This is confirmed by the assays with *PPIA* and *ACTB* +*GAPDH* as reference genes where there is absolutely no difference between the healthy MA and TC groups (p > 0.999, Fig 4b and 4c). However, when *SDHA* and *GUSB* are used as reference genes, a trend towards a difference could be detected (Fig 4a). Although not always significant, there is a clear trend for the upregulation of *c-MYC* in the mucosa from TC in patients with cholesteatoma (Fig 4a–4c). This trend is most clear when *ACTB* and *GAPDH* were used for normalisation (Fig 4c). Combined with the lowest variation in target gene expression and lack of expression difference between the healthy mucosa samples this validates *ACTB* and *GAPDH* as the most suitable reference genes for analysis of middle ear mucosa.

## Choice of control tissue affects the results with regard to TLR2 and TLR 4 expression

The validated qPCR method, including stable reference genes, was used to evaluate how the choice of control tissue affects qPCR results. QPCR is, by its nature, semi-quantitative and therefore the selection of good reference tissue is vital. In this study we have selected two different types of healthy tissue for evaluation as suitable reference tissue; MA and TC were collected from patients undergoing unrelated surgery, during cochlear implant (CI) or translabyrinthine vestibular schwannoma surgery.

No significant differences between MA or TC from healthy control, compared to mucosa from TC in cholesteatoma patients, regarding mRNA levels of *TLR2* could be observed

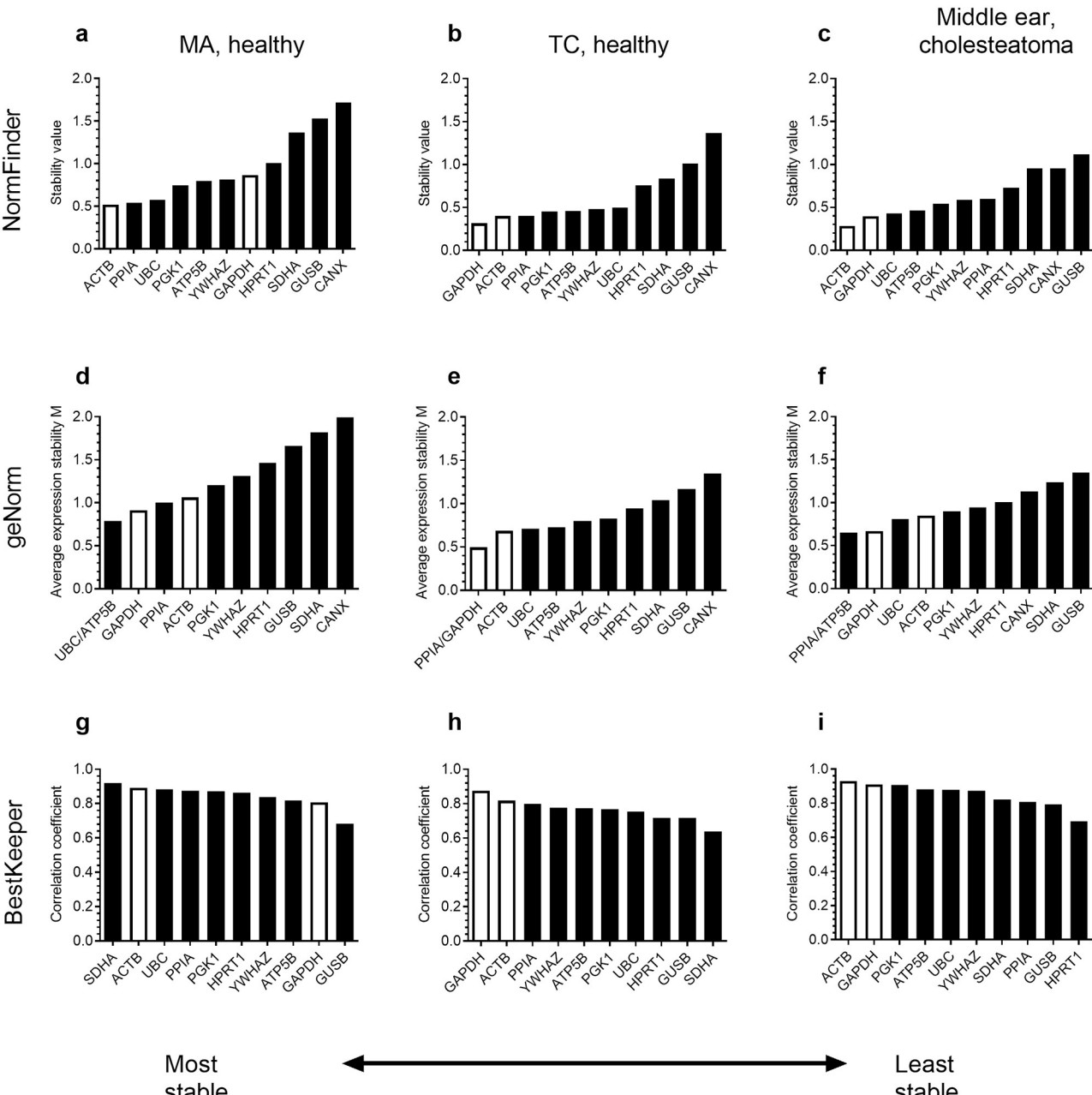

**Fig 2. Stability of reference genes.** The stability of the reference genes based on NormFinder (a-c), geNorm (d-f) and BestKeeper (g-i) analysis for mastoid antrum (MA) mucosa from healthy controls (n = 34) (a, d, g), tympanic cavity (TC) mucosa from healthy controls (n = 25) (b, e, h) and TC mucosa from cholesteatoma patients (n = 28) (c, f, i).

(Fig 5a). The levels of *TLR4* were not stable between the two different control tissues, and there was a strong trend (p = 0.065) towards a downregulation between MA and TC. This results in different interpretations regarding the dysregulation of *TLR4* in the TC in patients with cholesteatoma. *TLR4* is significantly downregulated in relation to MA, but there is no difference between TC from healthy controls and cholesteatoma patients (Fig 5b).

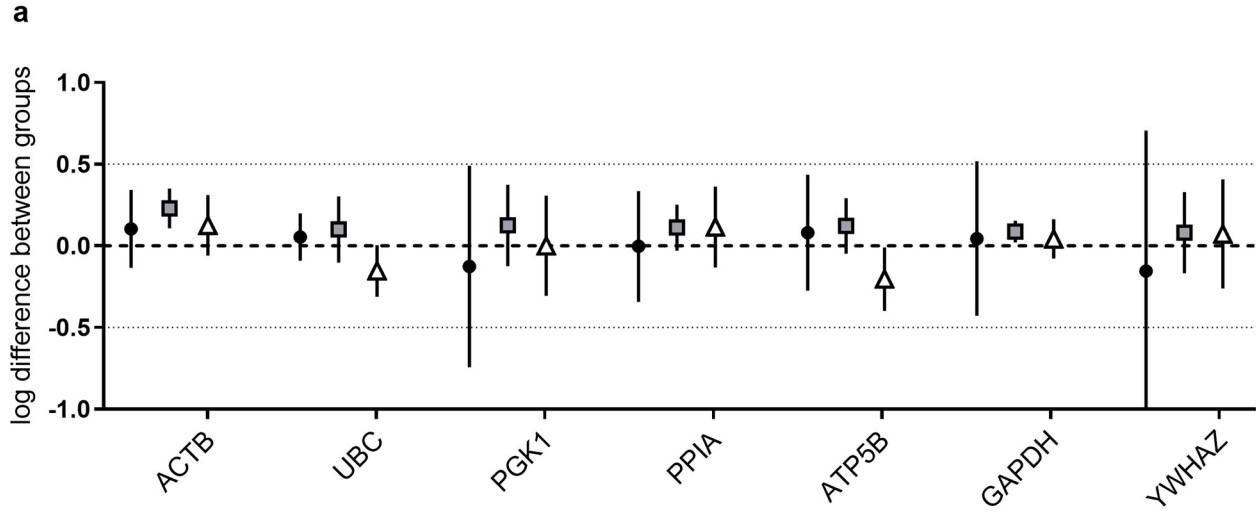

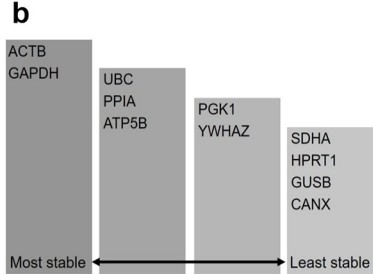

**Fig 3. Ranking of reference genes.** The top seven reference genes were analysed for their inter- and intragroup variation using NormFinder software (a). The intergroup variation is plotted as the expression difference between groups (● mastoid antrum (MA) mucosa from healthy controls (n = 34), □ tympanic cavity (TC) mucosa from healthy controls (n = 25), ▲ TC mucosa from cholesteatoma patients (n = 28)). The intragroup variation is indicated by vertical bars that give a confidence interval for the difference. Final ranking of the most and least stable reference genes for qPCR analysis of middle ear mucosa based on the analyses performed in this study (b).

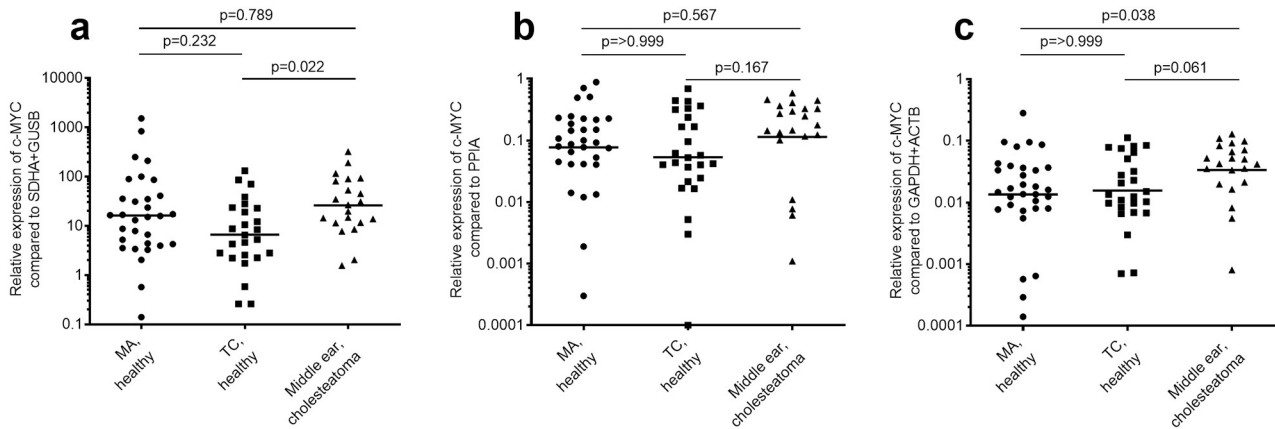

**Fig 4. Reference gene validation with *c-MYC* expression.** *C-MYC* expression was analysed using normalisation to *SDHA* and *GUSB* (a), *PPIA* (b) or *GAPDH* and *ACTB* (c). The *c-MYC* expression was compared between mastoid antrum (MA) mucosa from healthy controls (n = 32), tympanic cavity (TC) mucosa from healthy controls (n = 24) and TC mucosa from cholesteatoma patients (n = 21). After testing with the D'Agostino-Pearson test for Gaussian distribution the data was analysed with the one-way Kruskal-Wallis test with Dunn's post-test for statistical significance. The horizontal lines represent geometric means.

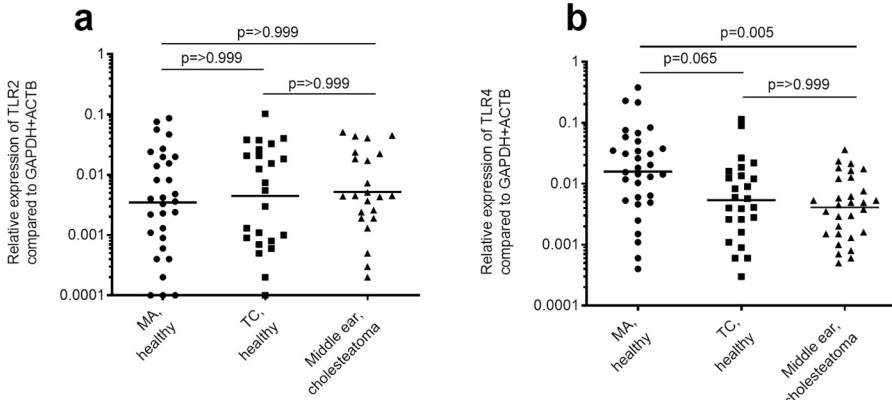

**Fig 5. Healthy control tissue evaluation with *TLR2* and *TLR4* expression.** *TLR2* (a) and *TLR4* (b) expression was analysed using normalisation to *GAPDH* and *ACTB*. The expression was compared between mastoid antrum (MA) mucosa from healthy controls ((a) n = 30 (b) n = 33), tympanic cavity (TC) mucosa from healthy controls ((a) n = 24 (b) n = 25) and TC mucosa from cholesteatoma patients ((a) n = 23 (b) n = 30). After testing with the D'Agostino-Pearson test for Gaussian distribution the data was analysed with the one-way Kruskal-Wallis test with Dunn's post-test for statistical significance. The horizontal lines represent geometric means.

## Discussion

The middle ear is a small and hard to reach compartment which has limited the possibilities for research in this area. The underlying molecular mechanisms in middle ear diseases, comprising cholesteatoma, are still largely unknown [8] and the possibilities for analysis of middle ear tissue is limited due to the fact that sampling of material could hazard the normal function of the ear. With the exception of Granath et al. and Hirai et al. where the expression of *TLR* is investigated in middle ear mucosa with regard to CMED in general, there is a lack of knowledge of the possible involvement of derangements of the mucosal innate immunity as a contributor to CMED and cholesteatoma [16, 17]. As previously mentioned, the amount of tissue possible to obtain from the middle ear mucosa is very small. The samples analysed in this study are paper-thin pieces about as large as the tip of a ballpoint pen. QPCR has proven to be a useful tool for the analysis of small tissue samples with great sensitivity and specificity [41]. Previously, the number of genes that could be studied was limited to the amount of RNA that could be extracted from the tissue sample. Also, small tissue amounts obstructed the analysis of genes with low mRNA expression levels. In recent years, the methods for amplification of whole-transcriptome cDNA have become reliable in the aspect of uniform amplification with minimal sequence bias [47]. This decreases the amount of sample required and allows for considerably more genes to be analysed per sample.

Not only is the amount of tissue limiting, but also the accessibility. Healthy human ear drums cannot be obtained due to ethical considerations. The mucosal layer in the MA and the TC originate from the same embryonic layer, the endoderm, and both consist of single layer of flattened to cuboidal respiratory epithelium [48]. Moreover, the MA is more readily accessible and can be obtained from patients undergoing surgery for non-middle ear related conditions, such as CI. Therefore, tissue from the MA could provide a suitable control for middle ear studies.

In this study, the dual-labelled probe method for qPCR was used. Since different fluorescent tags can be attached to the probe, multiple genes can be analysed simultaneously in the same well, provided a qPCR machine with multiple filters is available [49, 50]. The multiplex testing reduces the starting material required and reduces the variability introduced by technical

errors during PCR processing because the target gene and reference gene are amplified from the same starting material [49, 50]. The results presented in this paper show that multiplex analysis has comparable efficiency in regard to amplification compared to conventional single-plex qPCR. This opens up new possibilities for the research questions that can be answered with small tissue biopsies like those from the middle ear.

A very popular tool for reference gene identification is RefFinder [51] because it includes the three different algorithms from NormFinder, geNorm and BestKeeper to calculate expression stability. In this article RefFinder was not included because it does not take amplification efficiency into account. Transcription efficiency is important to include since not all genes are amplified equally and if not taken into account the methodological bias would be transferred to the results [41].

Since the three different software solutions are based on different formulas, they naturally give slightly different results. NormFinder has the advantage that it can take intergroup variation into account. This is important for the comparison between sample groups, which is the most common study design [42]. In this study, *GAPDH* was not always considered the most stable reference gene candidate within a sample group, but the intergroup variation showed that *GAPDH* is very stable between the different tissue groups. Therefore, *GAPDH* was ranked as a top candidate for the use as a reference gene in middle ear tissues.

A disadvantage with the geNorm algorithm is the risk of co-regulated genes being considered the most stably expressed because this algorithm looks for similarity in expression patterns [43]. Therefore, it is important to choose reference genes that do not belong to the same function class to avoid the risk of co-regulation. *ACTB* codes for a cytoskeletal protein while *GAPDH* encodes an enzyme involved in glycolysis and gluconeogenesis. As these are independent processes the risk for co-regulation between the two genes is minimal.

NormFinder and geNorm both have the disadvantage of not being able to handle missing data points [42, 43]. This means that every patient sample where one or more reference genes did not amplify has to be completely excluded for all potential reference genes. In this study, enough samples were collected in each group to be able to perform the analysis according to the software developer's recommendations. BestKeeper does not have this limitation but has the disadvantage of only being able to compare a maximum of ten genes [44]. In this study this was solved by performing this analysis last, enabling the exclusion of the potential reference gene considered least stable (*CANX*) in NormFinder and geNorm. The different properties of the various programs prove the importance of not relying on one software solution alone.

In the last decade it has become obvious that there is not one gene suitable for use of normalisation for all tissue types [40]. In the field of middle ear research many different genes have been used, such as *B2M* [52], *PPIA* [9], *ACTB* [33, 34], *HPRT1* [35] and *GAPDH* [36–39]. Interestingly, all articles published to this date use only one gene for normalisation, even though it has been abundantly shown that this is not sufficient for reliable analysis [40]. To our knowledge this is the first study to determine appropriate reference genes for normalisation in multiplex qPCR analysis of middle ear tissue. *ACTB* and *GAPDH* were shown to be the optimal genes for the normalisation of target gene expression when comparing healthy and diseased middle ear mucosa.

Even when the chosen reference genes are stably expressed, it is important to validate them. The optimal option is to use the expression of a gene of interest that is known to be altered in the target tissue. However, this was not possible in this project, since the mucosa in TC from cholesteatoma patients has not been studied before. *C-MYC* was chosen because it has previously been shown to be upregulated in cholesteatoma matrix with the use of *PPIA* for normalisation [9]. Since *c-MYC* is evolutionary stably expressed in several different cell-lines, and participates in a range of cellular functions such as proliferation and differentiation [53], it was

likely to be seen in the middle ear mucosa in healthy and cholesteatoma diseased ears. The experiments performed in this paper showed a clear upregulation of *c-MYC* with the use of the most stable reference genes (*PPIA*, *ACTB+GAPDH*). The same analysis with the least stable reference genes (*SDHA+GUSB*) showed no difference in *c-MYC* expression between control mucosa and mucosa from cholesteatoma ears. This indicates that the choice of reference genes is crucial for the detection of alternate expression of the target gene. Also, it is clear that the choice of reference genes affects the sensitivity of the analysis as demonstrated by the difference in variation between patient samples.

Despite surgical treatment there is a risk for recurrent cholesteatomas. At a molecular level, there are still a lot of unknown tasks concerning the cholesteatoma etiopathogenesis. Previous studies have shown dysregulation of *TLR2* and *TLR4* within the cholesteatoma matrix itself [18, 19, 27, 28]. Our focus was instead on the innate immunity of the mucosa in the middle ear, to search for its possible contribution to the cholesteatoma pathogenesis. An eardrum retraction as a sequelae to secretory otitis predisposes for a cholesteatoma development [3]. This makes it logic to speculate in the participation of factors belonging to the middle ear environment, and the risk to develop pathologic conditions, such as cholesteatoma. To get reliable results from qPCR data the selection of correct control tissue is essential. The method opens the possibility to analyse middle ear mucosa with the suggestion of a suitable control tissue. As highlighted by the differences in *TLR4* levels, depending on if MA or TC is used as a reference, the choice can affect the entire outcome of a study. We could not observe any clear dysregulation of *TLR2* or *TLR4* mRNA in the mucosa from patients with cholesteatoma.

The *c-MYC* results together with the *TLR2* and *TLR4* data illustrate the importance of the selection of both correct reference genes as well as control tissue. It can be argued that TC is the better control tissue to compare with other middle ear samples, but unfortunately TC samples are very difficult to obtain from healthy controls. MA on the other hand can be collected without complications from patients receiving a cochlear implant and is therefore the more realistic alternative. Compared to skin samples, MA is probably the superior choice of control tissue with regards to both structure and location. However, this would need to be confirmed in a future study.

It is our belief that a consensus regarding reference genes and the use of more precise control tissue will contribute to the comparability and reproducibility of studies within the field. To our knowledge this is the first study illustrating differential expression of genes in middle ear mucosa from ears with cholesteatoma. The upregulation of the *c-MYC* gene suggests participation of the mucosa in the development of the cholesteatoma disease, which should be further investigated in future studies.

## Methods

### Tissue sample collection

Mucosa was collected from the MA in the area of the lateral semicircular canal from individuals with healthy middle ears and mastoids during surgery for CI or during translabyrinthine surgery for vestibular schwannomas. Mucosa was also collected from the TC from the translabyrinthine group and from ears with cholesteatoma during surgery. For the cholesteatoma patient samples, care was taken to not include any skin cells or perimatrix. All samples were collected under sterile conditions with microsurgical technique at Linköping University Hospital, Sweden, during February 2011 to December 2016. Four different middle ear surgeons participated in the sample collection. All surgical procedures were performed under general anaesthesia. Three doses of prophylactic intravenous antibiotic (cefuroxime) was given at a 4 or 8 hourly interval during surgery as part of the CI, translabyrinthine and cholesteatoma

surgical routine. The samples were immediately put into RNAlater (Qiagen) and within 7 days stored in -70 ˚C until brought to Karolinska Institutet Stockholm, where again stored in -70 ˚C until analysis. The patients, or their parents if under the age of 18, were invited orally and in writing in close connection to surgery to participation in the study. A written informed consent was gathered from all participating individuals and the guardian in cases where the individual is below 18. An ethical approval from the local Ethical Review Board in Linköping supported the study (DNR 2011/88-31) and all research was performed in accordance with the declaration of Helsinki.

Human nasal epithelial cells (HNECs) from 3 healthy donors were isolated by nasal brushing, as previously described [54, 55] A written informed consent was gathered from all participating individuals. An ethical approval from the local Ethical Review Board in Stockholm supported the study (DNR 2016/823-31/2).

## RNA extraction, cDNA synthesis and Pre-amplification

The tissue samples were transferred from the RNAlater solution to RLT plus buffer (Qiagen) supplemented with DTT. The samples were disrupted using a ceramic bead-homogenizer (Precellys 24, Bertin Instruments), followed by treatment with a QIAshredder spin column (Qiagen). RNA was extracted using the RNeasy Micro Plus Kit (Qiagen) according to manufacturer's instructions. RNA yield and purity was assessed with a Nanodrop 1000 (Thermo Fisher Scientific). Samples with a yield of $\geq 2$ ng/µl and a 260/280 ratio $\geq 1.7$ were used for downstream applications. The QuantiTect Whole Transcriptome Kit (Qiagen) was used for cDNA synthesis and amplification using the manufacturer's protocol for high-yield reaction. RNA from HNEC was isolated using the RNeasy Micro Plus Kit as described above and cDNA was synthesized with the Omniscript RT Kit (Qiagen) with 400 ng RNA per reaction according to manufacturer's instructions.

## qPCR

All qPCR reactions were performed using the dual-labelled probe technique with the Quanti-Fast Multiplex PCR Kit (Qiagen) on a Stratagene mx3000p machine (Agilent Technologies). All reactions were run with the following program: activation at 95 ˚C for 5 min followed by 40 cycles of two-step cycling with denaturation at 95 ˚C for 45 s and annealing/extension at 60 ˚C for 45 s. Primers and probes for reference genes were obtained from Sigma Aldrich (KiCq-Start Probe Assays). Primers and probe for *c-MYC*, *TLR2* and *TLR4* were designed using Primer3 [56, 57] and Primer-BLAST software [30] and produced as Pure & Simple Primers and Dual-Labelled Probes (Sigma Aldrich). For specifications on genes and primers see Tables 2 and 3. The efficiency of reference genes was determined by using a dilution series of HNEC cDNA as a template. Efficiencies were tested in both singleplex and multiplex to ensure multiplex compatibility (Table 3). For identification of suitable reference genes duplex qPCR was run using cDNA from biopsies at a 1:100 dilution, primers at a concentration of 500 nM and probes at 200 nM. For validation of the reference genes *c-MYC* was chosen as a target gene. *C-MYC* was run together with reference genes using optimized primer concentrations (Table 3) and biopsy cDNA diluted 1:100. The two different healthy control tissues, analysed with *TLR2* and *TLR4*, was run together with reference genes using optimized primer concentrations (Table 3) and biopsy cDNA diluted 1:100.

## Data analysis

Ct determination was performed using automatic calculation of the MxPro software (Agilent Technologies, https://www.agilent.com/en/product/real-time-pcr-(qpcr)/real-time-pcr-

(qpcr)-instruments/mx3000-mx3005p-real-time-pcr-system-software/mxpro-qpcr-software-232751) and these values were used for further analysis with NormFinder version 0.953 [42] (https://moma.dk/normfinder-software) geNorm version 3 [43] (https://genorm.cmgg.be/) and BestKeeper version 1 [44] (https://www.gene-quantification.de/bestkeeper.html).Samples with missing Ct values for one or more genes were excluded from the analysis due to limitations of the software packages. For NormFinder and geNorm the Ct values were linearized and adjusted for efficiency using the following formula: Efficiency$^{(minCt-sampleCt)}$ where minCt is the lowest Ct value for that specific gene. These values were used to calculate the stability value for each reference gene in both software solutions, and intragroup variation and intergroup variation in NormFinder. For BestKeeper raw Ct values and primer efficiencies were used to calculate the correlation coefficient. For validation of reference genes, the expression of *c-MYC* was analysed using normalisation to different reference genes. If more than one reference gene was used the efficiency of the combination of those reference genes was taken into account, using the geometric mean.

## Statistics

All datasets were analysed for Gaussian distribution using the D'Agostino-Pearson omnibus normality test. For those where Gaussian distribution could be assumed one-way ANOVA tests with Holm-Sidak post-test were used to assess significant differences. For those where Gaussian distribution could not be assumed one-way Kruskal-Wallis tests with Dunn's post-tests were used. All statistical analyses were performed with GraphPad Prism version 8 software for Windows (GraphPad Software, Inc.).

## Acknowledgments

We would like to acknowledge the surgeons Henrik Harder, Jonas Frodlund and Markus Peebo who helped with sample collection.

## Author Contributions

**Conceptualization:** Johanna Westerberg, Elina Mäki-Torkko, Lars Olaf Cardell.

**Data curation:** Cecilia Drakskog, Susanna Kumlien Georén.

**Formal analysis:** Cecilia Drakskog, Nele de Klerk, Johanna Westerberg.

**Funding acquisition:** Lars Olaf Cardell.

**Investigation:** Cecilia Drakskog, Nele de Klerk, Johanna Westerberg.

**Methodology:** Cecilia Drakskog, Nele de Klerk.

**Project administration:** Cecilia Drakskog, Susanna Kumlien Georén.

**Resources:** Lars Olaf Cardell.

**Software:** Cecilia Drakskog, Nele de Klerk.

**Supervision:** Susanna Kumlien Georén.

**Validation:** Cecilia Drakskog, Nele de Klerk.

**Writing – original draft:** Cecilia Drakskog, Nele de Klerk, Johanna Westerberg.

**Writing – review & editing:** Cecilia Drakskog, Nele de Klerk, Johanna Westerberg, Elina Mäki-Torkko, Susanna Kumlien Georén, Lars Olaf Cardell.

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
