## [Decision Letter · Decision Letter 0]

16 Jun 2020

PONE-D-20-16025

Extensive qPCR analysis reveals altered gene expression in middle ear mucosa from cholesteatoma patients

PLOS ONE

Dear Dr. Cardell,

Thank you for submitting your manuscript to PLOS ONE. After careful consideration, we feel that it has merit but does not fully meet PLOS ONE’s publication criteria as it currently stands. Therefore, we invite you to submit a revised version of the manuscript that addresses the points raised during the review process.

We look forward to receiving your revised manuscript.

Kind regards,

Rafael da Costa Monsanto, M.D.

Academic Editor

PLOS ONE

2. In your Methods section, please provide additional details regarding the  human nasal epithelial cells used in your study. Please include the source from which you obtained the cells, the catalog number if applicable, whether the cell line was verified, and if so, how it was verified. For more information on PLOS ONE's guidelines for research using cell lines, see https://journals.plos.org/plosone/s/submission-guidelines#loc-cell-lines

3. Thank you for inlcuindg your funding statement; "NO - The funders had no role in study design, data collection and analysis, decision to publish, or preparation of the manuscript."

Reviewers' comments:

Reviewer's Responses to Questions

**Comments to the Author**

1. Is the manuscript technically sound, and do the data support the conclusions?

Reviewer #1: Yes

Reviewer #2: Yes

2. Has the statistical analysis been performed appropriately and rigorously? 

Reviewer #1: Yes

Reviewer #2: Yes

3. Have the authors made all data underlying the findings in their manuscript fully available?

Reviewer #1: Yes

Reviewer #2: Yes

4. Is the manuscript presented in an intelligible fashion and written in standard English?

Reviewer #1: Yes

Reviewer #2: Yes

5. Review Comments to the Author

Reviewer #1: I read the manuscript entitled “Extensive qPCR analysis reveals altered gene expression. In middle ear mucosa from cholesteatoma patients” with great interest.

The authors aimed to develop a stable qPCR method with validated reference genes and determination a suitable control tissue for the investigation of different diseases within middle ear and investigate whether the middle ear mucosa in ears with cholesteatoma has altered gene expression. In relation to its control tissue by the examination of TLR2, TL4 and c-MYC.

Major issues:

Introduction

1. Page 4, lines 61-69: Authors will work with TLR 2 and TLR 4. How did the authors determine the TLR? Is there any justification in the literature beyond Hirai et al. study showing a higher expression of TLR in immunohistochemical analysis in tissue samples from five cholesteatoma patients? Please clarify and show more about the studies showing dysregulation of TLR2 and TLR4 within the cholesteatoma matrix (18,19,42)

2. Page 4, line 65: Some references should be provided.

3. Page 4, line 73: Some references should be provided.

4. Page 4, line 78: Some references should be provided.

Results

1. Page 6, lines 122-4: Is there differences between the surgical aspects from the cholesteatoma patient’s? Expansion (atticus, antrum, mastoid)? Chain of auditory ossicles (destroyed, intact)? Recurrence?

2. Page 6, lines 123: Why the authors uses just mucosa from the TC from cholesteatoma patient’s?

3. Page 12, lines 226-8: Why the authors prefered to placed studies related to the C-MYC in the results and not in the introduction?

Discussion

1. Page 16, line 327: Some references should be provided.

2. Page 17, line 333: Some references should be provided.

3. Page 17, line 336: Some references should be provided.

4. Page 17, line 339: Some references should be provided.

5. Page 17, line 343: Some references should be provided.

6. Page 18, line 361: Some references should be provided.

Minor issues:

Abstract:

1. “Because recurrence cholesteatomas occur even when a microscopic surgical eradication of the disease is performed” – I suggest re-phrasing, as this sentence is somewhat confusing.

Reviewer #2: The goal of the study was to determine which reference genes were suitable for studying ME mucosa to enable studies of epithelium in grossly unaffected ME tissue as a hypothesis for recurrence. Because recurrent cholesteatomas occur even when complete microscopic surgical eradication of the disease is performed, it is likely that properties of the middle ear mucosa in apparently normal appearing mucosa contribute to the cholesteatoma recurrence. The aims of this paper were to develop a stable qPCR method with validated reference genes and determine a suitable control tissue for the investigation of different diseases within the middle ear. Using this method, they aimed to investigate whether the middle ear mucosa in ears with cholesteatoma has altered gene expression in relation to its control tissue by the examination of TLR2, TLR4 and c-MYC. In this paper reference gene candidates were evaluated in the middle ear mucosa of cholesteatoma patients and two different control tissues. 92 samples were used for cDNA synthesis and qPCR analysis. The mastoid antral (MA) samples were obtained from patients during cochlear implant (Cl) or translabyrinthine vestibular schwannoma surgery (n = 37). The expression of the eleven candidate reference genes was analyzed in mucosal biopsies taken from the MA and the tympanic cavity (TC) of healthy controls, and from the TC of cholesteatoma 149 patients.

In order to validate the choice of suitable reference genes, the highest ranked and lowest ranked genes were used for normalization of the expression of c-MYC as it has been shown that c-MYC is upregulated in cholesteatoma samples compared to retroauricular skin samples. Further indication for the suitability of the reference genes is the target gene expression in the control tissues. Combined with the lowest variation in target gene expression and lack of expression difference between the healthy mucosa samples this validates ACTB and GAPDH as the most suitable reference genes for analysis of middle ear mucosa. In this study, GAPDH was not always considered the most stable reference gene candidate within a sample group, but the intergroup variation showed that GAPDH is very stable between the different tissue groups. Therefore, GAPDH was ranked as a top candidate for the use as a reference gene in middle ear tissues.

The authors point out that it has become known that there is not just one gene suitable for use of normalization for all tissue types. In the field of middle ear research many different genes have been used, such as B2M, PPIA , ACTB , HPRT1 and GAPDH . Articles published to date typically use only one gene for normalization. The authors emphasize that reference genes need to be studied for suitability for the specific tissue under investigation.

Limitations discussed include that even when the chosen reference genes are stably expressed, it is important to validate them. The optimal option is to use the expression of a gene of interest that is known to be altered in the target tissue. However, this was not possible in this project, since the mucosa in TC from cholesteatoma patients has not been studied before. C-MYC was chosen because it has previously been shown to be upregulated in cholesteatoma matrix with the use of PPIA for normalization. Another limitation is the small amount of tissue available for analysis due to the limited availability of tissue that can be removed from the mucosa of the antrum or tympanic cavity.

To the authors’ knowledge this is the first study illustrating differential expression of genes in middle ear mucosa from ears with cholesteatoma. The upregulation of the c-MYC gene suggests participation of the mucosa in the development of the cholesteatoma disease, which should be further investigated in future studies.

I have no recommendations for revision at this time. The paper is clearly written and the authors acknowledge the need for future work to identify genes of interest in the mucosa of cholesteatoma patients. Hopefully, the identification of the suitable reference genes will enable this work to proceed.

6. PLOS authors have the option to publish the peer review history of their article (what does this mean?). If published, this will include your full peer review and any attached files.

Reviewer #1: No

Reviewer #2: No

---

## [Author Response · Author response to Decision Letter 0]

19 Aug 2020

PONE-D-20-16025

Extensive qPCR analysis reveals altered gene expression in middle ear mucosa from cholesteatoma patients

PLOS ONE

Line numbers are adjusted for “Manuscript Drakskog”.

2. In your Methods section, please provide additional details regarding the human nasal epithelial cells used in your study. Please include the source from which you obtained the cells, the catalog number if applicable, whether the cell line was verified, and if so, how it was verified. For more information on PLOS ONE's guidelines for research using cell lines, see https://journals.plos.org/plosone/s/submission-guidelines#loc-cell-lines

Thankyou for pointing this out to us, we have now added information regarding the origin of the human nasal epithelial cells used in the method section (see lines: 470-471).

Reviewers' comments:

Reviewer's Responses to Questions

Comments to the Author

1. Is the manuscript technically sound, and do the data support the conclusions?

Reviewer #1: Yes

Reviewer #2: Yes

2. Has the statistical analysis been performed appropriately and rigorously? 

Reviewer #1: Yes

Reviewer #2: Yes

3. Have the authors made all data underlying the findings in their manuscript fully available?

Reviewer #1: Yes

Reviewer #2: Yes________________________________________

4. Is the manuscript presented in an intelligible fashion and written in standard English?

Reviewer #1: Yes

Reviewer #2: Yes

5. Review Comments to the Author

Reviewer #1: I read the manuscript entitled “Extensive qPCR analysis reveals altered gene expression. In middle ear mucosa from cholesteatoma patients” with great interest.

The authors aimed to develop a stable qPCR method with validated reference genes and determination a suitable control tissue for the investigation of different diseases within middle ear and investigate whether the middle ear mucosa in ears with cholesteatoma has altered gene expression. In relation to its control tissue by the examination of TLR2, TL4 and c-MYC.

Major issues:

Introduction

1. Page 4, lines 61-69: Authors will work with TLR 2 and TLR 4. How did the authors determine the TLR? Is there any justification in the literature beyond Hirai et al. study showing a higher expression of TLR in immunohistochemical analysis in tissue samples from five cholesteatoma patients? Please clarify and show more about the studies showing dysregulation of TLR2 and TLR4 within the cholesteatoma matrix (18,19,42)

Answer: 

Thank you for highlighting this question, we have included some background information of why we address our focus on TLR2 and 4. See introduction line 65-102. 

In short, our aim was to investigate the relationship between innate immunity (TLR2 and 4), and the middle ear mucosa in healthy ears and ears with cholesteatoma. The innate immune system is important in the defense of pathogens entering the middle ear. As ear infections and chronic secretory otitis media may lead to an ear drum retractions, and the risk of developing a cholesteatoma, we chose to analyse the presence of TLR2 and 4 in the middle ear mucosa. 

This is a complement to earlier studies of TLRs and cholesteatoma who mainly focus on the cholesteatoma matrix (and/or perimatrix.) 

2. Page 4, line 65: Some references should be provided.

Answer: 

The sentence (“There is a lack of knowledge about the expression of TLRs in ears with cholesteatoma in comparison to healthy tissue.” ) is changed to : Toll Like Receptors (TLRs), as part of the mucosal innate immune system, play an important role as the first line of defense against an invasion of pathogenic infectious agents [14, 15], but their possible role in the cholesteatoma pathogenesis still needs to be clarified [16-19]. See line 66-69.

There is also added information in the introduction. 

3. Page 4, line 73: Some references should be provided.

Answer: 

Regarding: The amount of tissue available is especially limited when studying seemingly healthy mucosa from TC, since only a very small amount can be safely removed from the patient without interfering with recovery.

We provided a reference to the claim that a limited amount of tissue can be extracted from the middle ear without causing side effects to the patients, as well as restricted access to relevant healthy control tissue. “There is a risk for fibrous adhesions or even adhesive otitis media in cases of greater mucosal injuries in the middle ear [29].” See line 110-112

4. Page 4, line 78: Some references should be provided.

Answer:

Regarding: Semi-quantitative PCR (qPCR) is a suitable method because multiple factors can be analysed using very small amounts of tissue with high sensitivity, specificity and reproducibility.

We have provided a reference in line 113-115 for this statement.

Results

1. Page 6, lines 122-4: Is there differences between the surgical aspects from the cholesteatoma patient’s? Expansion (atticus, antrum, mastoid)? Chain of auditory ossicles (destroyed, intact)? Recurrence?

Answer:

We agree that more information about the patients are needed and has therefor added a table with this (see Table 1 in revised manuscript).

2. Page 6, lines 123: Why the authors uses just mucosa from the TC from cholesteatoma patient’s? 

Answer:

Our intention was to compare middle ear mucosa in healthy compared to cholesteatoma diseased ears. As we searched for a possible deranged mucosal innate immunity within the middle ear /tympanic cavity, this was the chosen site for sampling of mucosa specimen. 

We added a clarification in line 162-164. 

3. Page 12, lines 226-8: Why the authors prefered to placed studies related to the C-MYC in the results and not in the introduction?

Answer: We thank the reviewer for this comment. We have referred to c-MYC in the introduction (see lines 52-58).

Discussion

1. Page 16, line 327: Some references should be provided.

A reference has now been provided.

2. Page 17, line 333: Some references should be provided.

A reference has now been provided.

3. Page 17, line 336: Some references should be provided.

A reference has now been provided.

4. Page 17, line 339: Some references should be provided.

A reference has now been provided.

5. Page 17, line 343: Some references should be provided.

A reference has now been provided.

6. Page 18, line 361: Some references should be provided.

Answer: 

The sentence is changed (Since c-MYC is evolutionary stably expressed in several different cell-lines, and participates in a range of cellular functions such as proliferation and differentiation [53], it was likely to be seen in the middle ear mucosa in healthy and cholesteatoma diseased ears.) see line 411-413.

Minor issues:

Abstract:

1. “Because recurrence cholesteatomas occur even when a microscopic surgical eradication of the disease is performed” – I suggest re-phrasing, as this sentence is somewhat confusing.

Answer: 

Yes, we agree with this. The abstract is changed and the sentence ended up being removed. 

Reviewer #2: The goal of the study was to determine which reference genes were suitable for studying ME mucosa to enable studies of epithelium in grossly unaffected ME tissue as a hypothesis for recurrence. Because recurrent cholesteatomas occur even when complete microscopic surgical eradication of the disease is performed, it is likely that properties of the middle ear mucosa in apparently normal appearing mucosa contribute to the cholesteatoma recurrence. The aims of this paper were to develop a stable qPCR method with validated reference genes and determine a suitable control tissue for the investigation of different diseases within the middle ear. Using this method, they aimed to investigate whether the middle ear mucosa in ears with cholesteatoma has altered gene expression in relation to its control tissue by the examination of TLR2, TLR4 and c-MYC. In this paper reference gene candidates were evaluated in the middle ear mucosa of cholesteatoma patients and two different control tissues. 92 samples were used for cDNA synthesis and qPCR analysis. The mastoid antral (MA) samples were obtained from patients during cochlear implant (Cl) or translabyrinthine vestibular schwannoma surgery (n = 37). The expression of the eleven candidate reference genes was analyzed in mucosal biopsies taken from the MA and the tympanic cavity (TC) of healthy controls, and from the TC of cholesteatoma 149 patients.

In order to validate the choice of suitable reference genes, the highest ranked and lowest ranked genes were used for normalization of the expression of c-MYC as it has been shown that c-MYC is upregulated in cholesteatoma samples compared to retroauricular skin samples. Further indication for the suitability of the reference genes is the target gene expression in the control tissues. Combined with the lowest variation in target gene expression and lack of expression difference between the healthy mucosa samples this validates ACTB and GAPDH as the most suitable reference genes for analysis of middle ear mucosa. In this study, GAPDH was not always considered the most stable reference gene candidate within a sample group, but the intergroup variation showed that GAPDH is very stable between the different tissue groups. Therefore, GAPDH was ranked as a top candidate for the use as a reference gene in middle ear tissues.

The authors point out that it has become known that there is not just one gene suitable for use of normalization for all tissue types. In the field of middle ear research many different genes have been used, such as B2M, PPIA , ACTB , HPRT1 and GAPDH . Articles published to date typically use only one gene for normalization. The authors emphasize that reference genes need to be studied for suitability for the specific tissue under investigation.

Limitations discussed include that even when the chosen reference genes are stably expressed, it is important to validate them. The optimal option is to use the expression of a gene of interest that is known to be altered in the target tissue. However, this was not possible in this project, since the mucosa in TC from cholesteatoma patients has not been studied before. C-MYC was chosen because it has previously been shown to be upregulated in cholesteatoma matrix with the use of PPIA for normalization. Another limitation is the small amount of tissue available for analysis due to the limited availability of tissue that can be removed from the mucosa of the antrum or tympanic cavity.

To the authors’ knowledge this is the first study illustrating differential expression of genes in middle ear mucosa from ears with cholesteatoma. The upregulation of the c-MYC gene suggests participation of the mucosa in the development of the cholesteatoma disease, which should be further investigated in future studies.

I have no recommendations for revision at this time. The paper is clearly written and the authors acknowledge the need for future work to identify genes of interest in the mucosa of cholesteatoma patients. Hopefully, the identification of the suitable reference genes will enable this work to proceed.

---

## [Decision Letter · Decision Letter 1]

1 Sep 2020

Extensive qPCR analysis reveals altered gene expression in middle ear mucosa from cholesteatoma patients

PONE-D-20-16025R1

Dear Dr. Cardell,

We’re pleased to inform you that your manuscript has been judged scientifically suitable for publication and will be formally accepted for publication once it meets all outstanding technical requirements.

Kind regards,

Rafael da Costa Monsanto, M.D.

Academic Editor

PLOS ONE

Additional Editor Comments (optional):

Congratulations on the excellent work. 

Reviewers' comments:

Reviewer's Responses to Questions

**Comments to the Author**

1. If the authors have adequately addressed your comments raised in a previous round of review and you feel that this manuscript is now acceptable for publication, you may indicate that here to bypass the “Comments to the Author” section, enter your conflict of interest statement in the “Confidential to Editor” section, and submit your "Accept" recommendation.

Reviewer #2: All comments have been addressed

2. Is the manuscript technically sound, and do the data support the conclusions?

Reviewer #2: (No Response)

3. Has the statistical analysis been performed appropriately and rigorously? 

Reviewer #2: (No Response)

4. Have the authors made all data underlying the findings in their manuscript fully available?

Reviewer #2: (No Response)

5. Is the manuscript presented in an intelligible fashion and written in standard English?

Reviewer #2: (No Response)

6. Review Comments to the Author

Reviewer #2: (No Response)

7. PLOS authors have the option to publish the peer review history of their article (what does this mean?). If published, this will include your full peer review and any attached files.

Reviewer #2: No

---

## [Editor Report · Acceptance letter]

4 Sep 2020

PONE-D-20-16025R1 

Extensive qPCR analysis reveals altered gene expression in middle ear mucosa from cholesteatoma patients 

Dear Dr. Cardell:

I'm pleased to inform you that your manuscript has been deemed suitable for publication in PLOS ONE. Congratulations! Your manuscript is now with our production department. 

Kind regards, 

on behalf of

Dr. Rafael da Costa Monsanto 

Academic Editor

PLOS ONE